# CMB Tensions with Low-Redshift $H_0$ and $S_8$ Measurements: Impact of a Redshift-Dependent Type-Ia Supernovae Intrinsic Luminosity

**Matteo Martinelli** [1,*] **and Isaac Tutusaus** [2,3,4,*]

1    Institute Lorentz, Leiden University, PO Box 9506, 2300 RA Leiden, The Netherlands
2    Institute of Space Sciences (ICE, CSIC), Campus UAB, Carrer de Can Magrans, s/n, 08193 Barcelona, Spain
3    Institut d'Estudis Espacials de Catalunya (IEEC), Carrer Gran Capità 2-4, 08193 Barcelona, Spain
4    CNRS, IRAP, UPS-OMP, Université de Toulouse, 14 Avenue Edouard Belin, F-31400 Toulouse, France
*    Correspondence: martinelli@lorentz.leidenuniv.nl (M.M.); tutusaus@ice.csic.es (I.T.)

**Abstract:** With the recent increase in precision of our cosmological datasets, measurements of $\Lambda$CDM model parameter provided by high- and low-redshift observations started to be in tension, i.e., the obtained values of such parameters were shown to be significantly different in a statistical sense. In this work we tackle the tension on the value of the Hubble parameter, $H_0$, and the weighted amplitude of matter fluctuations, $S_8$, obtained from local or low-redshift measurements and from cosmic microwave background (CMB) observations. We combine the main approaches previously used in the literature by extending the cosmological model and accounting for extra systematic uncertainties. With such analysis we aim at exploring non standard cosmological models, implying deviation from a cosmological constant driven acceleration of the Universe expansion, in the presence of additional uncertainties in measurements. In more detail, we reconstruct the Dark Energy equation of state as a function of redshift, while we study the impact of type-Ia supernovae (SNIa) redshift-dependent astrophysical systematic effects on these tensions. We consider a SNIa intrinsic luminosity dependence on redshift due to the star formation rate in its environment, or the metallicity of the progenitor. We find that the $H_0$ and $S_8$ tensions can be significantly alleviated, or even removed, if we account for varying Dark Energy for SNIa and CMB data. However, the tensions remain when we add baryon acoustic oscillations (BAO) data into the analysis, even after the addition of extra SNIa systematic uncertainties. This points towards the need of either new physics beyond late-time Dark Energy, or other unaccounted systematic effects (particulary in BAO measurements), to fully solve the present tensions.

**Keywords:** cosmological observations; cosmological parameters; cosmic microwave background; type-Ia supernovae; cosmological tensions

---

## 1. Introduction

Since the beginning of modern cosmology, the value of the Hubble constant, $H_0$, providing the expansion rate of the universe today, has been one of the most important parameters in cosmology. The reason being that this quantity is used to construct time and distance cosmological scales. One of the first estimates of its value was provided by Hubble in 1929, $H_0 \sim 500\,\mathrm{km\,s^{-1}\,Mpc^{-1}}$ [1]. Its first measurement, $H_0 \sim 625\,\mathrm{km\,s^{-1}\,Mpc^{-1}}$, was eventually provided in 1927 by Lemaître [2]. Nearly 100 years later its value is believed to be significantly smaller and close to $70\,\mathrm{km\,s^{-1}\,Mpc^{-1}}$ but there is still no consensus on the exact number or its precision. The current methods to estimate the value of the Hubble constant can be roughly classified into two categories: local universe and

early universe estimates. In both cases there are assumptions that need to be made concerning the cosmological or astrophysical model assumed, but there may also remain systematic uncertainties that could bias the estimated value.

Let us first focus on the estimate of $H_0$ from the local universe. Most methods are essentially model-independent from a cosmological perspective; however, there are different ways to calibrate type-Ia supernovae (SNIa) distances or low-redshift probes that can be used to infer the expansion rate at present time. Using median statistics, some studies claim that $H_0$ should be equal to $68.0 \pm 5.5$ km s$^{-1}$ Mpc$^{-1}$ [3–5]. Since that time, many other analyses have claimed that they obtain values for $H_0$ close to 68 km s$^{-1}$ Mpc$^{-1}$ using different methods and assumptions. For instance, the authors in [6] determined the value of $H_0$ without using SNIa data and relying on measurements of the Hubble rate and their extrapolation to redshift zero. Other studies added baryon acoustic oscillations (BAO) and SNIa data into the analyses (see e.g., [7,8]). However, all these methods rely on an extrapolation to redshift zero. Instead, another way to estimate $H_0$ consists on measuring the distance to close Cepheids and use them to anchor our SNIa data. This allows us to build a distance ladder and infer the value of $H_0$. The SH0ES team [9] works on estimating the value of the Hubble constant using this method. The latest value provided by this team from recent Hubble Space Telescope observations of Cepheids in the Large Magellanic Cloud is equal to $74.03 \pm 1.42$ km s$^{-1}$ Mpc$^{-1}$ [10]. There have been many attempts at understanding whether or not this difference between local (low-redshift) measurements of the Hubble constant may be due to astrophysical systematic uncertainties not taken into account (see e.g., [10–17]), but there is no clear indication for a missing systematic uncertainty in the current estimate. It is also important to add that there have recently been analyses using strong lensing data which are consistent with the distance ladder estimates [18].

On the other hand, we can also estimate the value of $H_0$ using information from the early universe. Cosmic Microwave Background (CMB) surveys constrain the sound horizon at the last scattering surface ($\theta_*$) from which, assuming a model for the expansion history of the universe up to present time, the Hubble rate can be extrapolated. Using the concordance cosmological model, flat $\Lambda$CDM (Please note that in this work we consider the flat $\Lambda$CDM model as the concordance cosmological model. We will omit the reference to flatness in the following and just refer to it as $\Lambda$CDM), the best estimate we have presently obtained with Planck measurements of the CMB is equal to $67.36 \pm 0.54$ km s$^{-1}$ Mpc$^{-1}$ [19]. Please note that the estimate of $H_0$ obtained with the inverse distance ladder technique [20], where a standard pre-recombination physics is assumed and BAO data are used, is also consistent with the value obtained with CMB data alone. Many studies have been dedicated to solve the tension between late- and early-time estimates of $H_0$ both from a theoretical [21–39] and observational [10,16,40–44] point of view.

While the one on $H_0$ is indeed the most striking and statistically significant tension between current data sets, other inconsistencies between high- and low-redshift data have been found in recent years. In this paper, we will investigate also the tension in the matter clustering parameter $S_8$, which combines the matter density $\Omega_m$ and the amplitude of perturbations encoded in $\sigma_8$. This parameter has been measured by galaxy surveys, e.g., by the KiDS collaboration [45], and it has been found to be in tension with the value extrapolated by Planck measurements by $2.3\,\sigma$. While other surveys, such as DES [46] or HSC [47], found a less significant tension, also this discrepancy might be ascribed to either systematic effects in our measurements or to a failure of the standard $\Lambda$CDM model. Investigations of the first possibility have highlighted possible internal inconsistencies of the KiDS data (see e.g., [48]), while several theoretical models have been tested in an attempt to ease this tension (see e.g., [49,50]).

In this work we consider both possible sources, systematic effects and alternative cosmologies, for the tension on $H_0$ and $S_8$ between the local (or low-redshift) estimate and the high-redshift estimate obtained with CMB measurements; i.e., we consider possible astrophysical systematic effects in SNIa data and dark energy models beyond a cosmological constant. In more detail, we estimate the value of

the Hubble constant and $S_8$ using measurements at both ends of the cosmic time and we compare it with the local estimate using the distance ladder (for $H_0$) and low-redshift data (for $S_8$). We consider different astrophysical systematic effects that induce a redshift dependence on SNIa intrinsic luminosity, and, at the same time, different possible expansion histories: a cosmological constant, a dark energy fluid with constant equation of state (EoS) parameter, and a dark energy fluid with a model-independent EoS, with the latter approach arising from the necessity of testing model-independent expansion histories without having to rely on specific models or parametrizations [51–54].

This paper is organized as follows, in Section 2 we present the cosmological probes and the data sets used in this analysis, and in Section 3 we describe the methodology used to constrain the cosmological models. In Section 4 we show the results obtained for the different cosmological and astrophysical systematic effects models, and we discuss the tension on $H_0$ and $S_8$ in Section 5. We finish in Section 6 presenting our conclusions.

## 2. Cosmological Probes

In this section, we present the cosmological probes used in this analysis. When describing the supernovae data sets used, we pay special attention to the possible systematic effects that could lead to a redshift dependence of the inferred intrinsic luminosity.

### 2.1. Cosmic Microwave Background

The main aim of this paper is to assess how generalized cosmic expansion histories, together with additional SNIa systematic effects, can affect the significance of the tension between low- and high-redshift data. As our baseline high-redshift dataset we choose Planck 2015 [55], considering in our analysis the *TT-TE-EE* dataset together with the large-scale data from the *lowTEB* data. These data yield, when analyzed in a ΛCDM framework, an expansion rate $H_0 = 67.27 \pm 0.66$, a result which has a tension with local measurements $T(H_0) \approx 4\,\sigma$ [10]. At the same times, Planck data can be extrapolated to obtain a constraint on the amount of matter clustering $S_8 = 0.8331$ [19], a result also slightly in tension with low-redshift measurements, with a significance $T(S_8) = 2\,\sigma$ with respect to the results of KiDS [45], $T(S_8) \approx 1\,\sigma$ with the DES results [46], and no tension with the HSC results [47]. It was found that allowing for a more general expansion history, using the CPL parameterization [56,57], CMB prefers phantom equation of states for dark energy ($w(z) < -1$) with a significant worsening of the constraints on the expansion rate, while also significantly easing the tension with low-redshift measurements of the clustering of matter [49], even though such a tension reappears when limiting the expansion histories investigated to those produced by physically viable single field quintessence models [58].

### 2.2. Type-Ia Supernovae

Type-Ia supernovae are astrophysical objects considered standardizable candles which are useful to measure cosmological distances and break degeneracies present in other cosmological probes. The standard observable used in SNIa analyses is the so-called distance modulus,

$$\mu(z) = 5 \log_{10} \left( \frac{H_0}{c} d_{\mathrm{L}}(z) \right) , \tag{1}$$

where $d_{\mathrm{L}}(z) = (1 + z)r(z)$ is the luminosity distance, $r(z)$ the comoving distance, and $c$ the speed of light in vacuum. In the following we describe the standard treatment of SNIa observations for cosmological analyses, as well as different systematic effects that may introduce a redshift dependence in SNIa intrinsic luminosity.

#### 2.2.1. Standard Analysis

The standardization of SNIa is based on the empirical observation that these objects form a homogeneous class whose variability in their peak luminosity can be characterized by the stretch

of the light curve ($X_1$) and the color of the supernova at maximum brightness ($C$) [59]. Under the assumption that different SNIa with identical color, shape (of the light curve), and galactic environment have on average the same intrinsic luminosity for all redshifts, the observed distance modulus can be expressed as

$$\mu_{\mathrm{obs}} = m_{\mathrm{B}}^* - (M_{\mathrm{B}} - \alpha X_1 + \beta C), \tag{2}$$

where $m_{\mathrm{B}}^*$ stands for the observed peak magnitude in the B-band rest-frame, while $\alpha$, $\beta$, and $M_{\mathrm{B}}$ are nuisance parameters that need to be determined from the fit of our model to observations. They correspond to the amplitude of the stretch correction, the amplitude of the color correction, and the absolute magnitude of SNIa in the B-band rest-frame, respectively.

More recently, it has been shown [60,61] that $\beta$ and $M_{\mathrm{B}}$ depend on properties of the SNIa host galaxy. However, the mechanism for such dependence is not fully understood yet. In [62] the authors corrected for these dependencies assuming that the absolute magnitude $M_{\mathrm{B}}$ is related to the stellar mass of the host galaxy, $M_{\mathrm{stellar}}$ by a step function:

$$M_{\mathrm{B}} = \begin{cases} M_{\mathrm{B}}^1 & \text{if } M_{\mathrm{stellar}} < 10^{10} M_{\odot}, \\ M_{\mathrm{B}}^1 + \Delta_{\mathrm{M}} & \text{otherwise}, \end{cases} \tag{3}$$

where $M_{\mathrm{B}}^1$ and $\Delta_{\mathrm{M}}$ are two extra nuisance parameters that need to be determined from the fit, and $M_{\odot}$ corresponds to the mass of the Sun. Concerning $\beta$, the same authors claim that its dependence on the host stellar mass is too small to have a significant impact on cosmological analyses, and therefore can be neglected.

In this work we consider two different compilations of SNIa measurements: the joint light curve analysis (JLA) from [62], and the Pantheon compilation from [63]. Starting with JLA, it consists of the joint analysis of SNIa observations obtained from the three years of the SDSS survey together with observations from SNLS, HST, and several nearby experiments [64]. This provides a compilation of a total of 740 SNIa spanning from $z \approx 0.01$ to $z \approx 1$. The standardization used in JLA is the one presented in Equation (2) (see [62] for the technical details related to the fit of the light curves), and in this work we use the full covariance of the measurements provided by the authors. Several statistical and systematic uncertainties have been taken into account to determine this covariance, such as the error propagation of the light curve fit uncertainties, calibration, light curve model, bias correction, mass step, dust extinction, peculiar velocities, and contamination of nontype-IA supernovae. It is important to mention that this covariance matrix depends explicitly on the $\alpha$ and $\beta$ nuisance parameters. Therefore, we recompute the covariance matrix at each step when we sample the parameter space, and marginalize over $\alpha$ and $\beta$ to obtain constraints on cosmological parameters.

The other compilation considered in this work, Pantheon, contains SNIa measurements from the Pan-STARSS1 Medium Deep Survey, SDSS, SNLS, HST, and various low-redshift surveys. Pantheon is the largest compilation of SNIa measurements with a total amount of 1048 SNIa from $z \approx 0.01$ to $z \approx 2.3$. Besides the increased number of SNIa and the extension in redshift compared to JLA, the standardization of SNIa measurements is also slightly different. For instance, the mass step $\Delta_{\mathrm{M}}$, the stretch amplitude $\alpha$, and the color amplitude $\beta$ nuisance parameters are here pre-solved in a cosmology independent manner (see [63] for the details of the method). Therefore, the Pantheon compilation provides only the redshifts, distance moduli, and their covariance matrix. A detailed comparison between the different treatment of statistic and systematic uncertainties between these two compilations is beyond the scope of this work. We limit ourselves here to study the impact of these two compilations when determining cosmological constraints.

### 2.2.2. Redshift-Dependent Systematic Effects

Although our knowledge of the mechanism of SNIa detonation has significantly improved over the past few decades, there are several astrophysical systematic effects that still need to be understood (see e.g., [65] and references therein). Moreover, the difficulty in observing the system before it

becomes a SNIa (and therefore constrain our theoretical model for its detonation), as well as the difficulty of observing SNIa inside a very complex environment with multiple astrophysical processes that are hard to model, leaves enough uncertainty to deserve the consideration of whether or not a redshift dependence of the intrinsic luminosity of SNIa can have an impact on the cosmological conclusions we draw from them. Given the current and, in particular, the future precision of SNIa measurements [63,66,67] it is of major importance to understand if redshift-dependent systematic uncertainties need to be added in our cosmological analyses in order not to bias our final results (see e.g., [68,69] where non-accelerated models are shown to be able to fit the main cosmological probes, given enough redshift dependence of SNIa intrinsic luminosity). Evolution of the intrinsic luminosity of SNIa could appear also because of particular theoretical models. For instance, a varying gravitational constant, or a fine structure constant variation [70], would imply such redshift dependence. However, here we focus only on astrophysical origins for this kind of systematic uncertainties.

In this work we consider two different models for the redshift evolution of SNIa intrinsic luminosity: we first assume that SNIa intrinsic luminosity depends on the star formation rate (SFR) of its environment, while in the second case we assume that it depends on the metallicity of the environment.

Luminosity Dependence on the Star Formation Rate

Starting with the SFR model, several studies claim (see [14,15,71,72] and references therein (Please note that other studies claim that there is no significance for such an effect, like in [73] and references therein.)) that SNIa in younger environments are fainter (at more than $5\sigma$) than those in older environments after the standard light curve standardization. They also claim that this effect is still present if this environment dependence is added into the standardization together with the stretch, color, and mass step corrections. Since environmental ages evolve as a function of redshift, this dependence on the environment directly introduces an intrinsic luminosity dependence on redshift. More in detail, we know that the specific SFR (sSFR), the SFR normalized by the stellar mass, strongly depends on redshift (it decreases by an order of magnitude when going from $z = 1.5$ to $z = 0$ (see e.g., [74])). Theoretical predictions tell us that the sSFR is proportional to $(1+z)^{2.25}$ [75], while observations suggest that this dependence is even stronger: sSFR $\propto (1+z)^{2.8\pm0.2}$ [76]. Let us assume that the rate of young progenitors of SNIa is proportional to the SFR while the rate of old progenitors is proportional to the stellar mass of the host galaxy [77,78]. Then, the ratio between young and old progenitors would be proportional to the sSFR, and the measurement of the sSFR in regions in the vicinity of individual SNIa (local sSFR or LsSFR) would reflect this ratio in the surroundings of each SNIa. Let us denote the evolving fraction of young (old) SNIa progenitors as $\delta(z)$ ($\psi(z)$), respectively. The redshift evolution of their ratio is then given by

$$\frac{\delta(z)}{\psi(z)} \equiv \text{LsSFR}(z) = K \times (1+z)^{\phi}, \tag{4}$$

where $K$ is a constant that takes into account the approximation of replacing the sSFR by the LsSFR. It is important to mention that we implicitly assume that there is no survey selection efficiency against young or old progenitors. Although this is not perfectly true in real data, we consider this simplification here to provide a first quantitative estimate of the impact of this redshift dependence, while a detailed analysis with all the survey selection systematic effects is left for future work.

Given that $\delta(z) + \psi(z)$ must be equal to 1, we can write:

$$\delta(z) = (K^{-1} \cdot (1+z)^{-\phi} + 1)^{-1},$$
$$\psi(z) = (K \cdot (1+z)^{\phi} + 1)^{-1}. \tag{5}$$

According to the authors of [15], the value of $K$ should be roughly 0.87 to get a 50-50 split between old and young SNIa progenitors in their SNIa sample when using $\phi = 2.8$.

Let us further assume that the brightness offset between young and old populations, $\Delta_Y$, is constant with respect to redshift. This is what one would expect if this effect arises from the physics of the progenitors. The LsSFR could depend on the mean age of the old population. However, this would imply that $\Delta_Y$ would decrease as a function of redshift, since stars at higher redshift are younger, and this would amplify cosmological biases. Therefore, in this work we follow a conservative approach and assume $\Delta_Y$ to be constant. Under this assumption, the mean standardized magnitude of SNIa can be written as

$$\langle M_B^{\text{corr}}\rangle(z) = \delta(z) \times \langle M_B^{\text{corr}}\rangle_{\text{young}} + \psi(z) \times \langle M_B^{\text{corr}}\rangle_{\text{old}}$$
$$= \langle M_B^{\text{corr}}\rangle_{\text{young}} - \psi(z) \times \Delta_Y, \tag{6}$$

where the super-index corr indicates that the color, stretch, and mass step corrections are included.

We note that all the redshift dependence in Equation (6) has been encapsulated into the second term; therefore, we can combine this equation with Equation (2) by replacing $\langle M_B^{\text{corr}}\rangle_{\text{young}}$ by the standard color, stretch, and mass step standardization. This provides the final standardized distance modulus used in this work:

$$\mu_{\text{obs}} = m_B^* - (M_B - \alpha X_1 + \beta C + \psi(z) \times \Delta_Y). \tag{7}$$

The redshift evolution of the relation between the mass step and the sSFR corrections is complex [79–81]. However, the authors in [15] showed that even if the LsSFR correction can account for most of the luminosity dependence on the host galaxy, there is still roughly 30% of contribution from the mass step correction. Therefore, in this work we consider both corrections at the same time.

In practice, when using the SFR model for the luminosity dependence on redshift, we consider the following set of nuisance parameters in our analysis:

$$\{\alpha, \beta, M_B^1, \Delta_M, \Delta_Y, K, \phi\}, \tag{8}$$

with a Gaussian prior centered at 0.87 and width 0.2 for $K$, a Gaussian prior centered at 2.8 and width 0.2 for $\phi$, and a flat prior between $-0.5$ and $0.5$ for $\Delta_Y$.

Luminosity Dependence on Metallicity

Let us now focus on the second model considered in this work that introduces an intrinsic SNIa luminosity dependence on the redshift. Theoretical predictions suggest that the metallicity of the progenitor system of a SNIa could play a role in its maximum luminosity. More in detail, the maximum luminosity depends on the initial abundances of carbon, nitrogen, oxygen, and iron of the white dwarf progenitor [82–84]. Recently, the authors in [85,86] considered a theoretically motivated dependence of the absolute magnitude as a function of metallicity:

$$M_B(Z) = M_{B,Z_\odot} - 2.5 \log_{10}\left[1 - 0.18\frac{Z}{Z_\odot}\left(1 - 0.10\frac{Z}{Z_\odot}\right)\right] - 0.191 \,\text{mag}, \tag{9}$$

where $Z_\odot$ stands for the Solar metallicity. They performed an observational study and found a correlation between SNIa absolute magnitudes and the oxygen abundances of the host galaxies, showing that luminosities are higher for SNIa in galaxies with lower metallicities.

Although this relation is specific to each SNIa, we can consider the mean metallicity and derive the redshift dependence introduced according to this model. In this work we follow the approach of [87]. The mean cosmic metallicity $Z_b$ is given by [74]

$$Z_b = y\frac{\rho_*}{\Omega_b\rho_{\text{crit},0}}, \tag{10}$$

where

$$\rho_*(z) = (1 - R) \int_z^\infty \xi \frac{dz'}{H(z')(1+z')} \, , \tag{11}$$

the critical energy density today is given by

$$\rho_{\text{crit},0} = \frac{3H_0^2}{8\pi G} \, , \tag{12}$$

and the SFR is given by

$$\xi(z) = 0.015 \frac{(1+z)^{2.7}}{1 + [(1+z)/2.9]^{5.6}} M_\odot \, \text{yr}^{-1} \, \text{Mpc}^{-3} \, . \tag{13}$$

The yield $y$ and the return rate $R$ are nuisance parameters that depend on the initial mass function. Substituting the metallicity of a specific SNIa in Equation (9) by the mean cosmic metallicity given in Equation (10), and combining with Equation (2) we obtain the observed distance modulus for this model:

$$\mu_{\text{obs}} = m_{\text{B}}^* - \left\{ M_{\text{B}} - \alpha X_1 + \beta C - 2.5 \log_{10} \left[ 1 - 0.18 \frac{Z_b}{Z_\odot} \left( 1 - 0.10 \frac{Z_b}{Z_\odot} \right) \right] - 0.191 \, \text{mag} \right\} . \tag{14}$$

In practice, when considering this model, we take into account the following nuisance parameters in our analysis:

$$\{\alpha, \beta, M_{\text{B}}^1, \Delta_{\text{M}}, R, y\} . \tag{15}$$

In this work we consider the values of the yield and return rate provided in [88] for a Salpeter [89], Chabrier [90], and Kroupa [91,92] initial mass functions. In more detail, we consider a Gaussian prior centered at $y = 0.042$ with width 0.020, and a Gaussian prior centered at $R = 0.359$ with width 0.071, which come from the mean and standard deviation of the values provided in [88] for the different initial mass functions. Please note that also in this case we consider the mass step and metallicity correction at the same time to account for other astrophysical systematic effects (beyond metallicity) that could generate such dependence on the host galaxy. Let us finally mention that we take into account the uncertainty in the mean cosmic metallicity at redshift zero in comparison to the data through the priors on $y$ and $R$.

### 2.3. Baryon Acoustic Oscillations

The baryon acoustic oscillations are the characteristic patterns that can be observed in the distribution of galaxies in the large-scale structure of the universe. They are characterized by the length of a standard ruler, $r_d$. In the concordance cosmological model, BAO originate from sound waves propagating in the early universe. Therefore, the BAO scale $r_d$ corresponds to the comoving sound horizon at the redshift of the baryon drag epoch,

$$r_d = r_s(z_{\text{drag}}) = \int_{z_{\text{drag}}}^\infty \frac{c_s(z) \, dz}{H(z)} \, , \tag{16}$$

where $z_{\text{drag}} \approx 1060$ and $c_s(z)$ is the speed of sound as a function of redshift, which can be expressed as

$$c_s(z) = \frac{c}{\sqrt{3(1 + R_b(z))}} \, , \tag{17}$$

with $R_b(z) = 3\rho_b/(4\rho_\gamma)$. In this latter expression $\rho_b$ stands for the baryon energy density, while $\rho_\gamma$ corresponds to the photon energy density.

In this work we consider both isotropic and anisotropic measurements of the BAO. The observable used for isotropic measurements is given by $D_V(z)/r_d$, where the distance scale $D_V(z)$ can be expressed as

$$D_V(z) = \left( r^2(z) \frac{cz}{H(z)} \right)^{1/3}.$$

(18)

For the anisotropic measurements, the observables used are $c/(H(z) \cdot r_d)$ and $r(z)/r_d$, corresponding to the transverse and radial directions, respectively.

We use the isotropic measurements provided by 6dFGS at $z = 0.106$ [93] and by SDSS–MGS at $z = 0.15$ [94], and the anisotropic final results of BOSS DR12 at $z = 0.38, 0.51, 0.61$ [95] with their covariance matrix.

## 3. Methodology

We analyze the datasets presented in the previous section comparing them with the theoretical predictions given by three different cosmological models, distinguished by their expansion history:

- $\Lambda$CDM where equation of state (EoS) parameter of the dark energy component is $w(z) = -1$.
- $w$CDM with an EoS still constant like in the previous case, but with $w$ free to assume values different from the $\Lambda$CDM limit.
- $w(z)$CDM, a general case in which the EoS is binned in redshift and reconstructed using a smoothed step function, following the approach of [96]. This choice allows the exploration in a general way of the expansion history preferred by the data. Here we limit ourselves to the exploration of low-redshift dark energy effects, thus we divide the $w(z)$ function in 4 redshift bins, with $z_i = [0.05, 0.43, 0.82, 1.5]$; note that in [96] the binning choice was motivated by the use of theoretical priors, enforcing a correlation between the values of $w(z)$ at different redshifts. Here we do not make use of such priors, and we lower the number of redshift bins for computational purposes, limiting our analysis to the redshift range of interest for SNIa data.
  The function is therefore reconstructed using the values and errors of the $w_i$ parameters in each bin found by our analysis, with the assumption that after the last bin in redshift, the EoS stays constant in the past, i.e., $w(z > z_4) = w_4$. We note that this reconstruction method leads to equivalent results as the ones that can be obtained using Gaussian Processes, as it was shown in [96].

We assume that the expansion histories deviating from the $\Lambda$CDM behavior are driven by an additional minimally coupled scalar field, which changes the background expansion of the universe without directly affecting the evolution of cosmological perturbations. In our most general case, $w(z)$CDM, we allow the EoS to cross the so-called phantom divide $w = -1$; in the case of a minimally coupled single scalar field, such a model would generally develop ghost instabilities. Alternatively, single field DE models could cross the phantom divide removing the assumption of a minimal coupling to gravity [97], or if there is kinetical braiding [98,99]. We assume here that the underlying model producing this expansion history is effectively stable, i.e., it develops instabilities on time scales longer than those of interest for the analysis, or that such instabilities are mitigated by the presence of other scalar field (see discussion in [58]).

Notice that the expansion histories encompassed in the parametrizations described above are not able to mimic theories in which the background expansion is modified at early times (before matter-radiation equality), which have been found to be able to solve the tension between high- and low-redshift estimates of $H_0$ [32,36,37]. It is, however, able, in principle, to mimic the expansion history predicted by more exotic models of late-time DE, not included in the standard quintessence class, which have also been found to be good candidates to ease the $H_0$ tensions (see e.g., [28]).

The analysis we perform in this paper does not include the possibility that the tensions we investigate could be eased by modifications of the theory of gravity. Such modifications have been extensively explored in previous works (see e.g., [19,49,100]), and they could provide a promising

framework to tackle the tensions on both the expansion rate and the clustering of matter. While our approach on the EoS could mimic the expansion histories produced by such theories, modifications of gravity also imply a modified evolution of cosmological perturbations, which are not included in our analysis and would require additional care in the use of CMB and BAO data as some $\Lambda$CDM assumptions are done in their analysis [101,102].

Once the cosmological model is defined, we compare its prediction with the data and sample the parameter space using the public MCMC sampler `CosmoMC` [103,104]. We sample the 6 parameters of the minimal (flat) $\Lambda$CDM model: the baryon and cold dark matter densities at present day, $\Omega_b h^2$ and $\Omega_c h^2$; the optical depth, $\tau$; the primordial power spectrum amplitude and tilt, $A_s$ and $n_s$, and the Hubble constant $H_0$. We consider 1 massive neutrino of mass 0.06 eV and 2 massless neutrinos. In addition we include, when needed, the parameters describing the dark energy models alternative to $\Lambda$CDM, i.e., the constant $w$ for $w$CDM and the binned values $w_i$ for $w(z)$CDM. For these parameters we use flat priors.

On top of these cosmological parameters, we also sample the parameters describing the impact of systematics on the luminosity distance of SNIa, with their priors motivated in Section 2.2.2:

- SFR systematics: in this case we sample the parameters $K$, $\phi$ and $\Delta_Y$, using a Gaussian prior on the first two, with mean 0.87 and $\sigma = 0.2$ for $K$, and mean 2.8 and $\sigma = 0.2$ for $\phi$, while for $\Delta_Y$ we use a flat prior with range $[-0.5, 0.5]$.
- metallicity systematics: for this systematics model, the additional parameters are $y$ and $R$, both sampled with a Gaussian prior centered in 0.042 and 0.359, and $\sigma$ set to 0.02 and 0.071 respectively.

We quantify the tension between the high- and low-redshift measurements using as an estimate

$$T(\theta) = \frac{|\theta_{\text{high}} - \theta_{\text{low}}|}{\sqrt{\sigma_{\text{high}}^2 + \sigma_{\text{low}}^2}}, \tag{19}$$

with $\theta$ the parameter considered and $\sigma$ its Gaussian error. Please note that even limiting this tension estimator to the single parameter of interest, we still take into account the effects of other parameters, as we marginalize over all the parameter space except for $\theta$; we simplify however the assessment of the tension assuming Gaussian posteriors, and neglecting the impact of priors, but it is largely enough to determine if our models can alleviate or solve the tension. We refer the reader to [105] for a detailed analysis on precisely quantifying tensions.

In the following, we will compare our results with the local measurement of $H_0$ coming from the SH0eS collaboration [10] and the low-redshift measurement of $S_8 = \sigma_8 \sqrt{\Omega_m}$ from KiDS [45]. While the first low-redshift measurement is independent from the assumed cosmological model, the same does not apply to $S_8$, and therefore we re-analyze KiDS data in our extended dark energy models, using the CosmoMC module publicly released by the collaboration (https://github.com/sjoudaki/kids450).

## 4. Results

In this section, we present the results obtained through our analysis on cosmological and systematics parameters, focusing on the different cases one by one. The comparison between the cosmological and systematics models is instead discussed in the following section. Moreover, we discuss here only the results of the full Planck+SNIa+BAO dataset, leaving the discussion of the separate effects of SNIa and BAO to the following section.

### 4.1. Standard SNIa Analysis

In Figure 1 we show the constraints on the derived parameters $\Omega_m$ and $H_0$ obtained when no systematic effect is included in the luminosity distance of SNIa, both for the JLA (left panel) and Pantheon (right panel) datasets. As expected, the constraints enlarge moving from the $\Lambda$CDM expansion history to the more general $w$CDM and $w(z)$CDM cases. It is possible to notice however

how the posterior is not shifted between the three background expansions considered, highlighting how the constraints on the parameters of these are compatible with ΛCDM. We find moreover that except for slightly tighter constraints in the Pantheon case, the two SNIa datasets produce results in agreement between each other. A more complete list of constraints, showing the results for all the primary cosmological parameters can be found in Appendix A in Table A1.

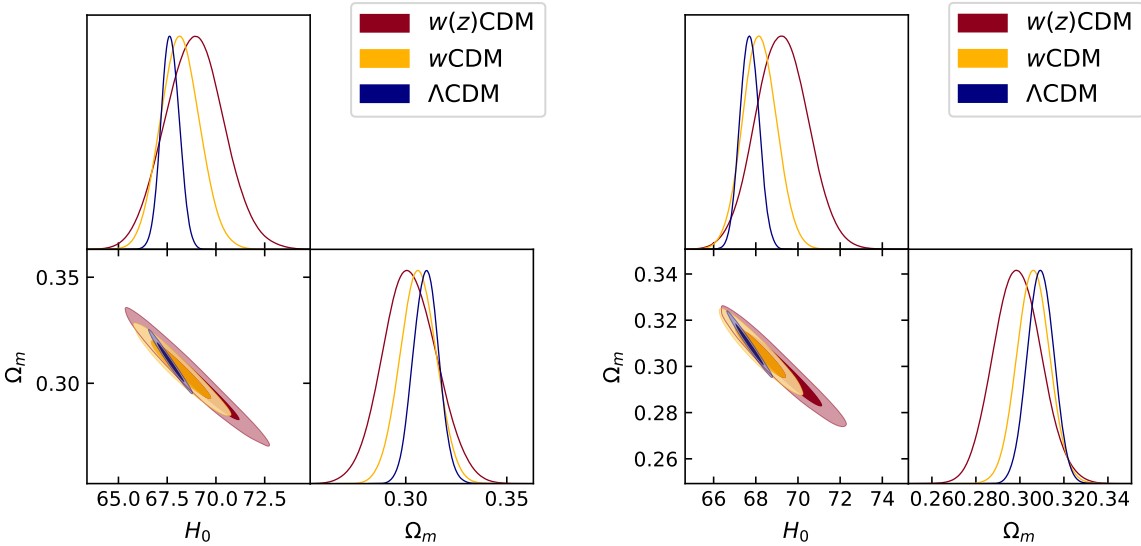

**Figure 1.** 68% and 95% confidence level contours for $\Omega_m$ and $H_0$ for the dark energy models explored (ΛCDM in blue, constant $w$ in yellow, and binned $w(z)$ in red) when no systematics are included in the analysis of the SNIa dataset. The data used are Planck + BAO + SNIa with SNIa datasets given by JLA (**left panel**) and Pantheon (**right panel**).

## 4.2. SNIa Luminosity Dependence on the Local Star Formation Rate

In Figure 2 the results shown refer to the case in which the luminosity of SNIa depends on the local star formation rate. We find again no significant difference on the cosmological parameter constraints when the different expansion histories are considered, except for the expected enlargement of the constraints.

Concerning the parameters controlling the systematic effect, we find that the constraints on $\phi$ and $K$ are dominated by the Gaussian prior we impose, while the amplitude parameter $\Delta_Y$, for which no Gaussian prior is added, is well constrained by the data around $\Delta_Y = 0$ (no systematic effects) in the ΛCDM and $w$CDM cases. When instead we reconstruct the EoS with the binned approach, $\Delta_Y$ shows a slight preference for negative values, with the $\Delta_Y = 0$ case still compatible at $1\sigma$.

Once again, the results on all cosmological and SNIa systematics parameters sampled in the analysis can be found in Table A2 of Appendix A.

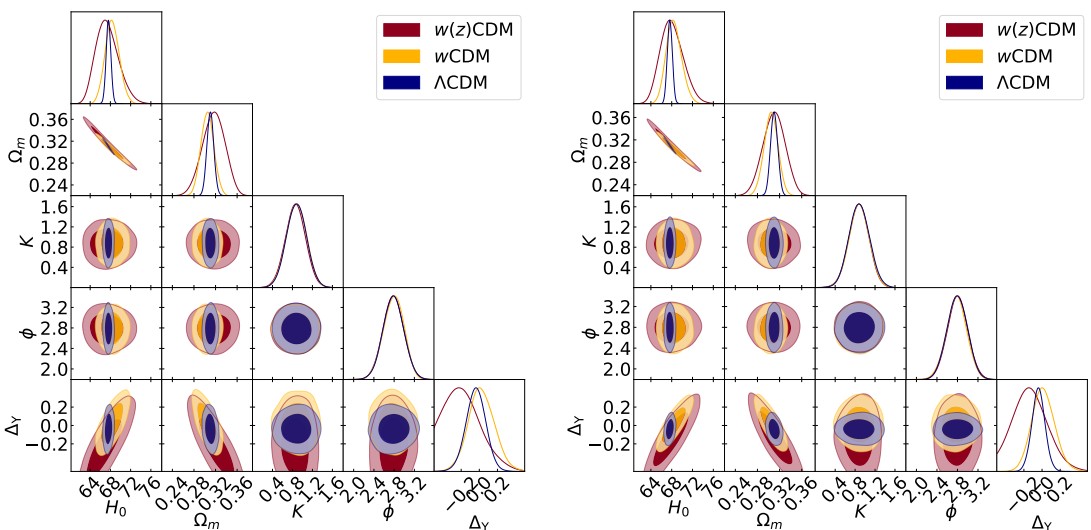

**Figure 2.** 68% and 95% confidence level contours for $\Omega_M$ and $H_0$ for the dark energy models explored ($\Lambda$CDM in blue, constant $w$ in yellow, and binned $w(z)$ in red) when star formation rate systematic effects are included in the analysis of the SNIa dataset. The data used are Planck+BAO+SNIa with SNIa datasets given by JLA (**left panel**) and Pantheon (**right panel**).

### 4.3. SNIa Luminosity Dependence on the Local Metallicity

The results obtained when the luminosity of SNIa depends on the environment metallicity are shown in Figure 3. Cosmological parameters constraints are listed in Table A3 of Appendix A, and they exhibit the same behavior as in the standard and SFR cases, with no shift in their posteriors when changing the dark energy EoS.

The systematics parameters in this case are $R$ and $y$, with both constraints dominated by the Gaussian prior.

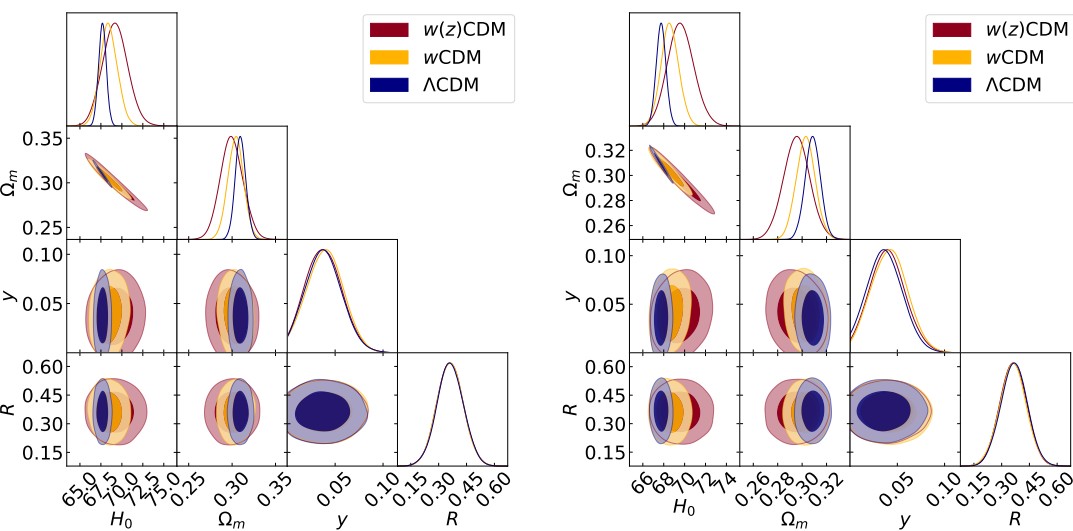

**Figure 3.** 68% and 95% confidence level contours for $\Omega_M$ and $H_0$ for the dark energy models explored ($\Lambda$CDM in blue, constant $w$ in yellow, and binned $w(z)$ in red) when metallicity systematic effects are included in the analysis of the SNIa dataset. The data used are Planck + BAO + SNIa with SNIa datasets given by JLA (**left panel**) and Pantheon (**right panel**).

## 5. Discussion: Generalized Expansion History and the High–Low-Redshift Tensions

In Section 4 we highlighted how the cosmological parameters do not shift their posterior distribution when the expansion history used to fit the data is changed, hinting for an agreement of the constraints on the Dark Energy EoS with $\Lambda$CDM. This can be seen clearly in Figure 4 where the constraints obtained on the $w(z)$ binned values are shown. In all 4 bins, the $w_i$ values are compatible with the $\Lambda$CDM limit at approximately $1\sigma$ with the most discrepant value found in the bin at the highest redshift. This is however affected by the fact that in our reconstruction we force $w(z)$ to be constant up to the recombination redshift after the last free bin. Therefore, the preference of CMB data for $w < -1$ [19,106] is the one driving this slight tension with $\Lambda$CDM.

We find therefore that even with the inclusion of possible systematic effects in the SNIa luminosity, no evidence for deviations from $\Lambda$CDM is found.

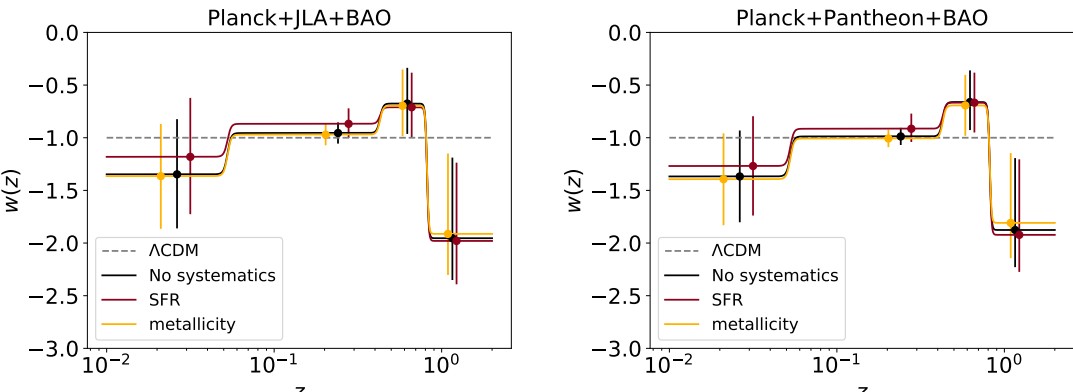

**Figure 4.** Reconstruction of the EoS in the $w(z)$CDM analysis. The points are the mean values of the posterior distributions obtained from the analysis, with the error bars corresponding to the 68% confidence limits. The results are shown for all the systematic models analyzed, i.e., SFR (red lines), metallicity (yellow line) and without any systematic effect (black lines). The data used are Planck + BAO + SNIa with SNIa datasets given by JLA (**left panel**) and Pantheon (**right panel**). Please note that the binned values of $w(z)$ are taken at the same redshifts in all three cases considered, with their spread in redshift artificially included only for better visualization.

We now turn our attention to the possibility that the generalized expansion histories that we consider together with the SNIa systematic effects might erase the tensions between CMB constraints and low-redshift measurements.

In Figure 5 and Table 1 we focus our attention to the tension between the $H_0$ value inferred from CMB data and the value obtained by the local measurements of the SH0eS collaboration [10] (gray band). We report here the results obtained with the Planck+SNIa+BAO combination (solid lines) together with the case in which we do not include BAO data (dashed line), with the left and right panels including JLA and Pantheon SNIa datasets respectively. For the Planck+SNIa case, we find that when the considered expansion history is the most general one, $w(z)$CDM, the tension is significantly eased, if not completely removed, in all systematic effects cases, with $T(H_0) \approx 0.3\,\sigma$ (see the specific tensions for these and the remaining cases in Table 1). For the less general $w$CDM dark energy model instead, we find no significant easing of the tension for the metallicity ($T(H_0) \approx 2.7\,\sigma$) and no systematic effects cases ($T(H_0) \approx 3\,\sigma$), while we are still able to find an agreement with SH0eS measurements if SFR effects are included ($T(H_0) \approx 0.9\,\sigma$). Finally, in $\Lambda$CDM, we find no significant effect of the possible redshift evolution of SNIa luminosity on the $H_0$ tension when the systematic effects are added ($T(H_0) \approx 4.3\,\sigma$).

**Table 1.** Tension between the high- and low-redshift measurements of $H_0$ and $\sigma_8\sqrt{\Omega_m}$.

| Parameter | Dark Energy Case | Planck + JLA | Planck + JLA + BAO | Planck + Pantheon | Planck + Pantheon + BAO |
|---|---|---|---|---|---|
| | | NO systematics | | | |
| $T(H_0)$ | $\Lambda$CDM | 4.3 | 4.3 | 4.3 | 4.3 |
| | $w$CDM | 2.7 | 3.4 | 3.3 | 3.6 |
| | $w(z)$CDM | 0.3 | 2.5 | 1.1 | 2.5 |
| $T(\sigma_8\sqrt{\Omega_m})$ | $\Lambda$CDM | 2.5 | 2.4 | 2.5 | 2.4 |
| | $w$CDM | 2.2 | 2.2 | 2.2 | 2.1 |
| | $w(z)$CDM | 2.3 | 2.6 | 2.4 | 2.6 |
| | | SFR systematics | | | |
| $T(H_0)$ | $\Lambda$CDM | 4.3 | 4.3 | 4.3 | 4.3 |
| | $w$CDM | 0.9 | 2.8 | 0.9 | 2.7 |
| | $w(z)$CDM | 0.1 | 2.5 | 0.2 | 2.3 |
| $T(\sigma_8\sqrt{\Omega_m})$ | $\Lambda$CDM | 2.5 | 2.4 | 2.5 | 2.4 |
| | $w$CDM | 2.0 | 2.1 | 2.0 | 2.1 |
| | $w(z)$CDM | 2.2 | 2.7 | 2.2 | 2.6 |
| | | metallicity systematics | | | |
| $T(H_0)$ | $\Lambda$CDM | 4.3 | 4.3 | 4.2 | 4.2 |
| | $w$CDM | 2.5 | 3.1 | 2.9 | 3.3 |
| | $w(z)$CDM | 0.1 | 2.3 | 0.8 | 2.3 |
| $T(\sigma_8\sqrt{\Omega_m})$ | $\Lambda$CDM | 2.5 | 2.4 | 2.4 | 2.3 |
| | $w$CDM | 2.2 | 2.1 | 2.2 | 2.2 |
| | $w(z)$CDM | 2.2 | 2.6 | 2.4 | 2.5 |

When BAO data are included, the tension is increased with respect to the Planck+SNIa data combination, with the inclusion of BAO dragging the results toward smaller $H_0$ values. In the $w(z)$CDM case, the inclusion of BAO in the analyzed dataset has its strongest effect in our third redshift bin, with $w_3$ lying below the phantom line ($w = -1$) for Planck+SNIa, but above it in the Planck+SNIa+BAO (see Appendix A). The general effect of this change is to drag the preferred $H_0$ back to small values, in tension with the local measurements at the level of $T(H_0) \approx 2.4\,\sigma$. This result points toward the BAO data as those preventing a generalized Dark Energy EoS to be able to solve the tension with local $H_0$ measurements. The BAO measurements affecting this redshift bin are those at the highest redshifts given by BOSS DR12 [95]. While possible systematic uncertainties on these measurements, or the removal of these high-redshift BAO data might allow the easing of tensions without the need to neglect BAO as a whole, a detailed analysis on the mechanisms that might drive this behavior are beyond the scope of this paper.

Figure 5 also reports (in green) the bound on $H_0$ obtained by Planck alone in the three dark energy models considered: note that Planck data alone provide very loose constraints on the equation of state parameter's trend in redshift, which yields the very high value of $H_0$ found both in $w$CDM and $w(z)$CDM, consistent with what is found by the Planck collaboration [106].

Another important, although less statistically significant, tension between high- and low-redshift measurements is the one on the estimate of the clustering of matter; this is usually encoded in the parameter $\sigma_8$, i.e., the amplitude of the (linear) power spectrum on scales of $8h^{-1}$ Mpc. Recent measurements from low-redshift galaxy surveys found results on the combined parameter $S_8 = \sigma_8\sqrt{\Omega_m}$ which differ from those inferred from CMB constraints. In Figure 6 we compare the results of the KiDS collaboration [45] (gray band) with the results obtained with the Planck+SNIa+BAO combination (solid lines) together with the case in which we do not include BAO data (dashed line), and the Planck alone case (green solid lines), again with the left and right panels referring to JLA and Pantheon SNIa datasets respectively.

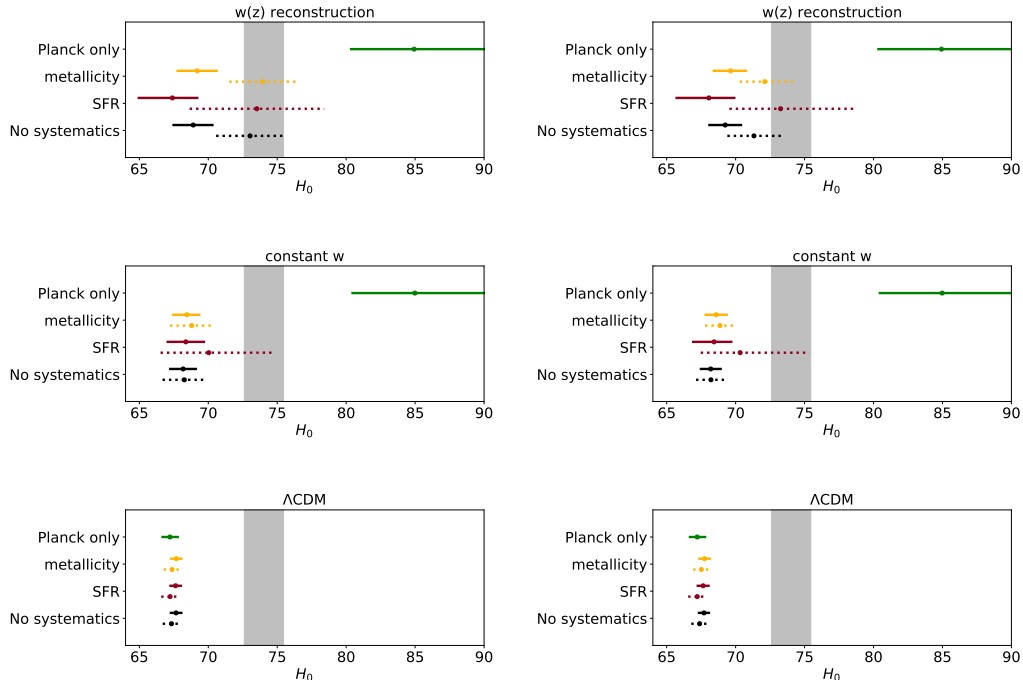

**Figure 5.** Visualization of the $H_0$ tension between Planck+SNIa and the local measurement. The error bars correspond to the 68% errors for the different cases explored in this paper, while the gray band highlights the $1\sigma$ bound of the SH0eS collaboration. The data used are Planck + BAO + SNIa with SNIa datasets given by JLA (**left panel**) and Pantheon (**right panel**).

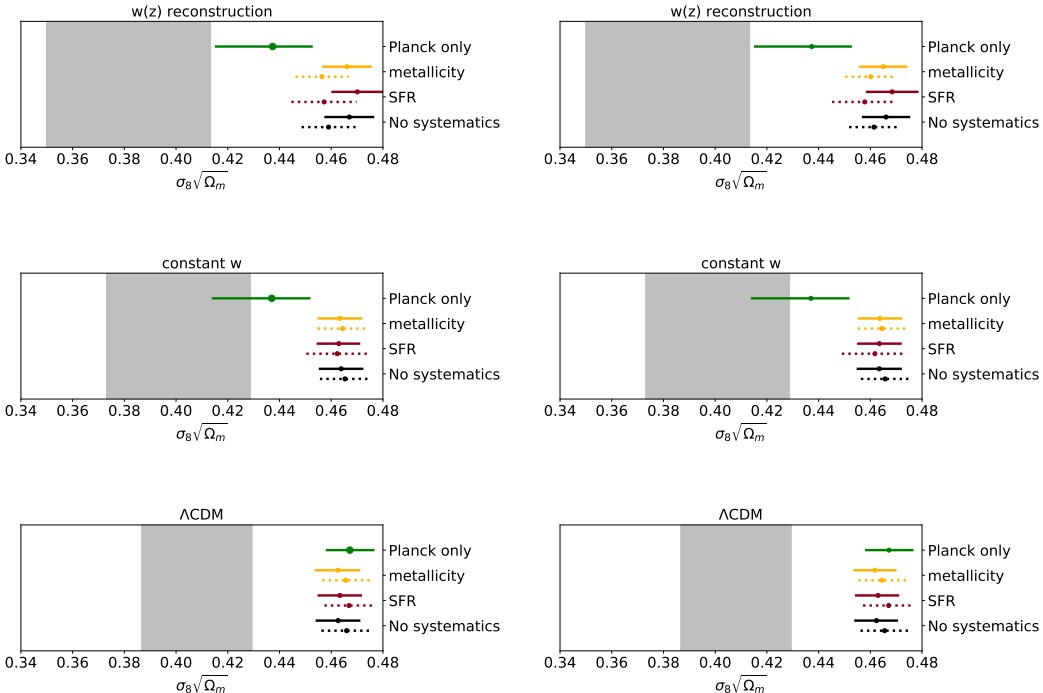

**Figure 6.** Visualization of the tension on $\sigma_8\sqrt{\Omega_m}$ between Planck+SNIa and the measurement obtained by KiDS Weak Lensing survey. The error bars correspond to the 68% errors for the different cases explored in this paper, while the gray band highlights the $1\sigma$ bound of the KiDS collaboration. The data used are Planck+BAO+SNIa with SNIa datasets given by JLA (**left panel**) and Pantheon (**right panel**).

We find that when Planck is not combined with other datasets, the tension is easily solved generalizing the expansion history with respect to $\Lambda$CDM, as the constraints on $\sigma_8\sqrt{\Omega_m}$ from Planck are compatible with those of KiDS both for $w$CDM and $w(z)$CDM, a result compatible with what is found by the KiDS collaboration [49].

Including the SNIa data sets forces $w$ to be closer to the $\Lambda$CDM limit, thus shifting back the results to higher values of $\sigma_8\sqrt{\Omega_m}$ with respect to the Planck only case. While for the $H_0$ tension, we found that SFR systematic effects allow the constraints in the $w$CDM and $w(z)$CDM cosmologies to be closer to the KiDS bound with respect to the metallicity and no systematic effects cases, this is not the case for $\sigma_8\sqrt{\Omega_m}$: we find no significant difference between the systematic models, with a tension $T(H_0) \approx 2\,\sigma$ for the two DE models.

Once again, including the BAO data shifts the constraints further away from the low-redshift results, as it happens for $H_0$.

## 6. Conclusions

In this paper, we investigated the possibility of easing the tensions between CMB and low-redshift measurements generalizing the expansion history at late times. Together with a standard $\Lambda$CDM evolution, we considered a $w$CDM case in which the EoS parameter can deviate from $w = -1$ but is still constant in redshift, and a reconstructed $w(z)$CDM where the EoS is reconstructed in four redshift bins. As CMB is not able by itself to provide precise information on the late-time evolution, we included SNIa and BAO data to tighten the constraints. We explored the possibility that SNIa suffer from unconsidered systematic effects, affecting their intrinsic luminosity as a function of redshift. We considered in this context, two possible redshift evolutions, one connected with the star formation rate in the environment of the SNIa, and one related to the metallicity of their progenitor system.

We discussed the constraints on cosmological parameters, highlighting how these are affected by both the generalization of the expansion history and the introduction of systematic effects. We focused in particular on the effect that these systematics have on the reconstruction of $w(z)$ and on the tensions with low-redshift measurements. In addition to this, we found that given our choice of priors on the systematic effect parameters, cosmological data are not able to provide tight constraints on them, and these are dominated by the priors (except for $\Delta_Y$). It could be interesting to extend the analysis leaving more freedom to these parameters; note however that the priors used in this work are observationally motivated, as different values for $R$, $y$, $K$, $\phi$ would lead to unrealistic trends in redshift of the star formation rate and metallicity.

We found that for all the datasets and systematic considered, the reconstruction of the EoS parameter is compatible with the $\Lambda$CDM limit $w = -1$ (see Figure 4) with differences between the systematic cases not statistically significant.

We then focused on the impact of the different analysis configurations on the easing of tensions on $H_0$ and $\sigma_8\sqrt{\Omega_m}$; we found that for the latter, both the generalized expansion and the systematic effects do not impact significantly the tension when using both the Planck + SNIa and Planck + SNIa + BAO datasets.

We found instead that the tension with the $H_0$ measurements from the SH0eS collaboration is significantly eased in the $w(z)$CDM dark energy model for all systematic cases when the Planck + SNIa data combination is considered, while for $w$CDM only the SFR systematic has a significant effect on the tension. Interestingly, we found that including the BAO data in the analysis drags the $H_0$ we obtain towards values that are again in tension with SH0eS. In the $w(z)$CDM case, this effect seems to arise from the fact that while in the Planck + SNIa combination the third reconstruction bin yields $w_3 < -1$, when BAO data are included the reconstruction favors $w_3 > -1$, while in all other redshift bins the results obtained in the Planck + SNIa and Planck + SNIa + BAO combinations are compatible with each other (see Tables A1–A3). This bin corresponds to the redshift range where the BOSS DR12 measurements lie; a more detailed investigation of what causes this shift in the $w(z)$ reconstruction is certainly needed, but we chose to leave this for a future work.

Overall, we found no significant impact of the systematic effects considered on the tensions between high- and low-redshift cosmological constraints, at least for the prior ranges on their parameters that we discussed in Section 2.2.2. We found instead that generalizing the late-time expansion history allows easing of this tension. However, while when considering CMB and SNIa data, the mean value of $H_0$ is actually shifted toward the local measurements values, when the BAO data are included the mean values are kept at low $H_0$ with only a slight increase of the errors with respect to the $\Lambda$CDM case. Such a result is in agreement with the previous analysis of [33], where the authors found that exotic evolution of the DE fluid allow significant easing of tensions between low- and high-redshift measurements, as long as SNIa and BAO are considered separately. It would be therefore of interest to assess the agreement between these two background probes to further investigate this effect.

Finally, we want to comment on the apparent inconsistency between our results and those of [29], where the authors found that a sharp transition in $w(z)$ for $1 < z < 2$ seems to ease the tension both on $\sigma_8$ and $H_0$ between high- and low-redshift measurements. In [29] the authors use of data at higher redshift with respect to our analysis, which allows them to reconstruct $w(z)$ to redshifts higher than our last bin at $z = 1.5$. The missing evidence for such a sharp transition in $w(z)$ can therefore be attributed to the different data set choice, a possibility we aim at investigating in a future work.

**Author Contributions:** For the development of this paper, MM contributed with the coding of the Dark Energy equation of state reconstruction and of the SNIa systematic effects into the analysis pipeline, and with the statistical analysis of the results, as well as the writing of the paper. IT contributed with the development of the theoretical framework for the SNIa systematic effects, and performing the data analysis, as well as with the writing of the paper.

**Funding:** This work has been carried out thanks to the support of the OCEVU Labex (ANR-11-LABX- 0060) and of the Excellence Initiative of Aix-Marseille University—A*MIDEX, part of the French "Investissements d'Avenir" programme. MM acknowledges support from the D-ITP consortium, a program of the NWO that is funded by the OCW.

**Acknowledgments:** We thank Alessandra Silvestri and Dan Scolnic for very fruitful comments and discussions which helped to noticeably improve this work. We also thank André Tilquin for his help with the use of the DEC cluster.

**Conflicts of Interest:** The authors declare no conflict of interest.

## Appendix A. Constraints on Cosmological Parameters

In this Appendix, we report the constraints on all the main cosmological parameters sampled in our analysis. In Table A1 we show the constraints obtained when no systematics effects are included, while Tables A2 and A3 contain, respectively, results when Star Formation Rate and metallicity systematic effects are considered.

**Table A1.** Marginalized values of the parameters and their 68% confidence level bounds, obtained using Planck+SNIa and Planck+SNIa+BAO, with no systematic effects included. When only upper or lower bounds are found, we report the 95% confidence level limit.

| Parameter | Dark Energy Case | Planck + JLA | Planck + JLA + BAO | Planck + Pantheon | Planck + Pantheon + BAO |
|---|---|---|---|---|---|
| $\Omega_b h^2$ | $\Lambda CDM$ | $0.02225 \pm 0.00016$ | $0.02230 \pm 0.00014$ | $0.02226 \pm 0.00015$ | $0.02231 \pm 0.00013$ |
| | $wCDM$ | $0.02223 \pm 0.00016$ | $0.02228 \pm 0.00015$ | $0.02223 \pm 0.00016$ | $0.02227 \pm 0.00015$ |
| | $w(z)CDM$ | $0.02226 \pm 0.00016$ | $0.02220 \pm 0.00016$ | $0.02224 \pm 0.00016$ | $0.02221 \pm 0.00015$ |
| $\Omega_c h^2$ | $\Lambda CDM$ | $0.1197 \pm 0.0014$ | $0.1189 \pm 0.0010$ | $0.1195 \pm 0.0014$ | $0.11884 \pm 0.00098$ |
| | $wCDM$ | $0.1200 \pm 0.0014$ | $0.1193 \pm 0.0013$ | $0.1200 \pm 0.0015$ | $0.1194 \pm 0.0012$ |
| | $w(z)CDM$ | $0.1198 \pm 0.0015$ | $0.1204 \pm 0.0014$ | $0.1200 \pm 0.0015$ | $0.1204 \pm 0.0014$ |
| $\tau$ | $\Lambda CDM$ | $0.080 \pm 0.017$ | $0.083 \pm 0.016$ | $0.081 \pm 0.017$ | $0.084 \pm 0.016$ |
| | $wCDM$ | $0.077 \pm 0.017$ | $0.082 \pm 0.017$ | $0.077 \pm 0.017$ | $0.081 \pm 0.017$ |
| | $w(z)CDM$ | $0.074 \pm 0.017$ | $0.072 \pm 0.018$ | $0.074 \pm 0.017$ | $0.073 \pm 0.017$ |
| $\log 10^{10} A_s$ | $\Lambda CDM$ | $3.094 \pm 0.033$ | $3.099 \pm 0.032$ | $3.096 \pm 0.033$ | $3.100 \pm 0.032$ |
| | $wCDM$ | $3.090 \pm 0.033$ | $3.098 \pm 0.033$ | $3.090 \pm 0.033$ | $3.095 \pm 0.033$ |
| | $w(z)CDM$ | $3.082 \pm 0.034$ | $3.081 \pm 0.034$ | $3.082 \pm 0.033$ | $3.082 \pm 0.033$ |

**Table A1.** *Cont.*

| Parameter | Dark Energy Case | Planck + JLA | Planck + JLA + BAO | Planck + Pantheon | Planck + Pantheon + BAO |
|---|---|---|---|---|---|
| | $\Lambda CDM$ | $0.9650 \pm 0.0048$ | $0.9668 \pm 0.0041$ | $0.9653 \pm 0.0046$ | $0.9671 \pm 0.0040$ |
| $n_s$ | $wCDM$ | $0.9643 \pm 0.0047$ | $0.9659 \pm 0.0043$ | $0.9641 \pm 0.0047$ | $0.9657 \pm 0.0044$ |
| | $w(z)CDM$ | $0.9646 \pm 0.0049$ | $0.9631 \pm 0.0047$ | $0.9641 \pm 0.0048$ | $0.9632 \pm 0.0047$ |
| | $\Lambda CDM$ | $-$ | $-$ | $-$ | $-$ |
| $w_1$ | $wCDM$ | $-1.036 \pm 0.053$ | $-1.023 \pm 0.042$ | $-1.035 \pm 0.037$ | $-1.025 \pm 0.034$ |
| | $w(z)CDM$ | $-1.37^{+0.56}_{-0.50}$ | $-1.35 \pm 0.51$ | $-1.34 \pm 0.45$ | $-1.37 \pm 0.44$ |
| | $\Lambda CDM$ | $-$ | $-$ | $-$ | $-$ |
| $w_2$ | $wCDM$ | $-$ | $-$ | $-$ | $-$ |
| | $w(z)CDM$ | $-0.83 \pm 0.12$ | $-0.96 \pm 0.10$ | $-0.922 \pm 0.097$ | $-0.988 \pm 0.082$ |
| | $\Lambda CDM$ | $-$ | $-$ | $-$ | $-$ |
| $w_3$ | $wCDM$ | $-$ | $-$ | $-$ | $-$ |
| | $w(z)CDM$ | $-1.51^{+0.60}_{-0.52}$ | $-0.68^{+0.34}_{-0.29}$ | $-1.04^{+0.42}_{-0.38}$ | $-0.66^{+0.30}_{-0.27}$ |
| | $\Lambda CDM$ | $-$ | $-$ | $-$ | $-$ |
| $w_4$ | $wCDM$ | $-$ | $-$ | $-$ | $-$ |
| | $w(z)CDM$ | $< -0.92$ | $-1.95^{+0.77}_{-0.40}$ | $-2.58^{+1.6}_{-0.97}$ | $-1.88^{+0.68}_{-0.35}$ |
| | $\Lambda CDM$ | $67.33 \pm 0.64$ | $67.66 \pm 0.47$ | $67.39 \pm 0.61$ | $67.70 \pm 0.45$ |
| $H_0$ | $wCDM$ | $68.3 \pm 1.6$ | $68.2 \pm 1.0$ | $68.2 \pm 1.1$ | $68.19 \pm 0.81$ |
| | $w(z)CDM$ | $73.0 \pm 2.5$ | $68.9 \pm 1.5$ | $71.3^{+2.2}_{-1.9}$ | $69.2 \pm 1.2$ |
| | $\Lambda CDM$ | $0.4660 \pm 0.0097$ | $0.4627 \pm 0.0086$ | $0.4655 \pm 0.0095$ | $0.4624 \pm 0.0085$ |
| $\sigma_8 \Omega_m^{1/2}$ | $wCDM$ | $0.4653 \pm 0.0097$ | $0.4639 \pm 0.0087$ | $0.4658 \pm 0.0095$ | $0.4635 \pm 0.0087$ |
| | $w(z)CDM$ | $0.459 \pm 0.011$ | $0.4670 \pm 0.0097$ | $0.4614 \pm 0.0098$ | $0.4661 \pm 0.0094$ |

**Table A2.** Marginalized values of the parameters and their 68% confidence level bounds, obtained using Planck+SNIa and Planck+SNIa+BAO, when SFR systematic effects are included. When only upper or lower bounds are found, we report the 95% confidence level limit.

| Parameter | Dark Energy Case | Planck + JLA | Planck + JLA + BAO | Planck + Pantheon | Planck + Pantheon + BAO |
|---|---|---|---|---|---|
| | $\Lambda CDM$ | $0.02223 \pm 0.00016$ | $0.02230 \pm 0.00014$ | $0.02223 \pm 0.00016$ | $0.02230 \pm 0.00014$ |
| $\Omega_b h^2$ | $wCDM$ | $0.02224 \pm 0.00015$ | $0.02227 \pm 0.00015$ | $0.02224 \pm 0.00016$ | $0.02227 \pm 0.00015$ |
| | $w(z)CDM$ | $0.02226 \pm 0.00016$ | $0.02221 \pm 0.00015$ | $0.02224 \pm 0.00016$ | $0.02222 \pm 0.00015$ |
| | $\Lambda CDM$ | $0.1199 \pm 0.0015$ | $0.1190 \pm 0.0010$ | $0.1199 \pm 0.0015$ | $0.1190 \pm 0.0011$ |
| $\Omega_c h^2$ | $wCDM$ | $0.1199 \pm 0.0015$ | $0.1194 \pm 0.0013$ | $0.1199 \pm 0.0015$ | $0.1195 \pm 0.0013$ |
| | $w(z)CDM$ | $0.1197 \pm 0.0015$ | $0.1204 \pm 0.0014$ | $0.1200 \pm 0.0014$ | $0.1203 \pm 0.0014$ |
| | $\Lambda CDM$ | $0.079 \pm 0.017$ | $0.083 \pm 0.016$ | $0.078 \pm 0.017$ | $0.083 \pm 0.017$ |
| $\tau$ | $wCDM$ | $0.078 \pm 0.016$ | $0.080 \pm 0.018$ | $0.077 \pm 0.017$ | $0.080 \pm 0.017$ |
| | $w(z)CDM$ | $0.073 \pm 0.018$ | $0.073 \pm 0.017$ | $0.073 \pm 0.017$ | $0.074 \pm 0.017$ |
| | $\Lambda CDM$ | $3.092 \pm 0.033$ | $3.098 \pm 0.032$ | $3.092 \pm 0.034$ | $3.100 \pm 0.033$ |
| $\log 10^{10} A_s$ | $wCDM$ | $3.091 \pm 0.032$ | $3.093 \pm 0.034$ | $3.090 \pm 0.033$ | $3.094 \pm 0.033$ |
| | $w(z)CDM$ | $3.080 \pm 0.034$ | $3.082 \pm 0.034$ | $3.081 \pm 0.033$ | $3.084 \pm 0.033$ |
| | $\Lambda CDM$ | $0.9643 \pm 0.0049$ | $0.9667 \pm 0.0041$ | $0.9643 \pm 0.0047$ | $0.9668 \pm 0.0041$ |
| $n_s$ | $wCDM$ | $0.9643 \pm 0.0047$ | $0.9655 \pm 0.0047$ | $0.9642 \pm 0.0047$ | $0.9654 \pm 0.0046$ |
| | $w(z)CDM$ | $0.9648 \pm 0.0048$ | $0.9632 \pm 0.0047$ | $0.9641 \pm 0.0047$ | $0.9634 \pm 0.0047$ |
| | $\Lambda CDM$ | $-$ | $-$ | $-$ | $-$ |
| $w_1$ | $wCDM$ | $-1.09^{+0.11}_{-0.16}$ | $-1.032^{+0.059}_{-0.053}$ | $-1.102^{+0.089}_{-0.16}$ | $-1.035^{+0.065}_{-0.053}$ |
| | $w(z)CDM$ | $-1.36 \pm 0.51$ | $-1.18 \pm 0.54$ | $-1.41 \pm 0.45$ | $-1.27 \pm 0.47$ |
| | $\Lambda CDM$ | $-$ | $-$ | $-$ | $-$ |
| $w_2$ | $wCDM$ | $-$ | $-$ | $-$ | $-$ |
| | $w(z)CDM$ | $-0.84^{+0.19}_{-0.21}$ | $-0.87^{+0.15}_{-0.13}$ | $-0.98^{+0.14}_{-0.20}$ | $-0.91^{+0.14}_{-0.13}$ |
| | $\Lambda CDM$ | $-$ | $-$ | $-$ | $-$ |
| $w_3$ | $wCDM$ | $-$ | $-$ | $-$ | $-$ |
| | $w(z)CDM$ | $-1.49^{+0.63}_{-0.54}$ | $-0.71^{+0.33}_{-0.28}$ | $-1.15^{+0.48}_{-0.42}$ | $-0.67 \pm 0.28$ |
| | $\Lambda CDM$ | $-$ | $-$ | $-$ | $-$ |
| $w_4$ | $wCDM$ | $-$ | $-$ | $-$ | $-$ |
| | $w(z)CDM$ | $< -1.02$ | $-1.98^{+0.74}_{-0.41}$ | $-2.6^{+1.7}_{-1.0}$ | $-1.92^{+0.72}_{-0.35}$ |
| | $\Lambda CDM$ | $67.23 \pm 0.65$ | $67.64 \pm 0.47$ | $67.21 \pm 0.65$ | $67.65 \pm 0.48$ |
| $H_0$ | $wCDM$ | $70.0^{+4.8}_{-3.5}$ | $68.4 \pm 1.4$ | $70.3^{+5.0}_{-2.9}$ | $68.4^{+1.3}_{-1.6}$ |
| | $w(z)CDM$ | $73.5 \pm 4.6$ | $67.4^{+1.9}_{-2.5}$ | $73.3^{+5.3}_{-3.7}$ | $68.1^{+1.9}_{-2.4}$ |

**Table A2.** *Cont.*

| Parameter | Dark Energy Case | Planck + JLA | Planck + JLA + BAO | Planck + Pantheon | Planck + Pantheon + BAO |
|---|---|---|---|---|---|
| $\sigma_8\Omega_m^{1/2}$ | $\Lambda CDM$ | $0.4669 \pm 0.0096$ | $0.4627 \pm 0.0088$ | $0.4671 \pm 0.0099$ | $0.4630 \pm 0.0087$ |
| | $wCDM$ | $0.462 \pm 0.012$ | $0.4630 \pm 0.0084$ | $0.462^{+0.011}_{-0.013}$ | $0.4635 \pm 0.0086$ |
| | $w(z)CDM$ | $0.457 \pm 0.012$ | $0.470 \pm 0.010$ | $0.458 \pm 0.012$ | $0.468 \pm 0.010$ |
| $K$ | $\Lambda CDM$ | $0.88 \pm 0.20$ | $0.87 \pm 0.20$ | $0.88 \pm 0.20$ | $0.88 \pm 0.21$ |
| | $wCDM$ | $0.89 \pm 0.20$ | $0.88 \pm 0.20$ | $0.88 \pm 0.20$ | $0.88 \pm 0.20$ |
| | $w(z)CDM$ | $0.87 \pm 0.20$ | $0.86 \pm 0.20$ | $0.87 \pm 0.20$ | $0.88 \pm 0.20$ |
| $\phi$ | $\Lambda CDM$ | $2.79 \pm 0.20$ | $2.78 \pm 0.20$ | $2.78 \pm 0.20$ | $2.80 \pm 0.20$ |
| | $wCDM$ | $2.81 \pm 0.21$ | $2.79 \pm 0.20$ | $2.80 \pm 0.20$ | $2.78 \pm 0.20$ |
| | $w(z)CDM$ | $2.79 \pm 0.20$ | $2.78 \pm 0.20$ | $2.81 \pm 0.20$ | $2.80 \pm 0.20$ |
| $\Delta_Y$ | $\Lambda CDM$ | $-0.06 \pm 0.11$ | $-0.03 \pm 0.10$ | $-0.059 \pm 0.076$ | $-0.040 \pm 0.072$ |
| | $wCDM$ | *unconstrained* | $0.02 \pm 0.14$ | $> -0.34$ | $0.01 \pm 0.13$ |
| | $w(z)CDM$ | *unconstrained* | $-0.18^{+0.14}_{-0.25}$ | *unconstrained* | $-0.12^{+0.17}_{-0.21}$ |

**Table A3.** Marginalized values of the parameters and their 68% confidence level bounds, obtained using Planck+SNIa and Planck+SNIa+BAO, when metallicity systematic effects are included. When only upper or lower bounds are found, we report the 95% confidence level limit.

| Parameter | Dark Energy Case | Planck + JLA | Planck + JLA + BAO | Planck + Pantheon | Planck + Pantheon + BAO |
|---|---|---|---|---|---|
| $\Omega_b h^2$ | $\Lambda CDM$ | $0.02226 \pm 0.00015$ | $0.02231 \pm 0.00014$ | $0.02228 \pm 0.00015$ | $0.02232 \pm 0.00014$ |
| | $wCDM$ | $0.02223 \pm 0.00016$ | $0.02227 \pm 0.00015$ | $0.02223 \pm 0.00015$ | $0.02226 \pm 0.00015$ |
| | $w(z)CDM$ | $0.02226 \pm 0.00016$ | $0.02220 \pm 0.00015$ | $0.02224 \pm 0.00016$ | $0.02221 \pm 0.00016$ |
| $\Omega_c h^2$ | $\Lambda CDM$ | $0.1196 \pm 0.0014$ | $0.1189 \pm 0.0010$ | $0.1193 \pm 0.0014$ | $0.1187 \pm 0.0010$ |
| | $wCDM$ | $0.1199 \pm 0.0015$ | $0.1195 \pm 0.0012$ | $0.1200 \pm 0.0015$ | $0.1196 \pm 0.0012$ |
| | $w(z)CDM$ | $0.1197 \pm 0.0015$ | $0.1204 \pm 0.0014$ | $0.1199 \pm 0.0015$ | $0.1203 \pm 0.0014$ |
| $\tau$ | $\Lambda CDM$ | $0.080 \pm 0.017$ | $0.084 \pm 0.016$ | $0.082 \pm 0.017$ | $0.084 \pm 0.017$ |
| | $wCDM$ | $0.078 \pm 0.017$ | $0.080 \pm 0.017$ | $0.077 \pm 0.017$ | $0.080 \pm 0.017$ |
| | $w(z)CDM$ | $0.073 \pm 0.017$ | $0.072 \pm 0.018$ | $0.074 \pm 0.017$ | $0.073 \pm 0.018$ |
| $\log 10^{10} A_s$ | $\Lambda CDM$ | $3.095 \pm 0.033$ | $3.100 \pm 0.032$ | $3.099 \pm 0.032$ | $3.101 \pm 0.033$ |
| | $wCDM$ | $3.091 \pm 0.034$ | $3.094 \pm 0.032$ | $3.090 \pm 0.033$ | $3.094 \pm 0.032$ |
| | $w(z)CDM$ | $3.081 \pm 0.033$ | $3.080 \pm 0.034$ | $3.083 \pm 0.033$ | $3.082 \pm 0.034$ |
| $n_s$ | $\Lambda CDM$ | $0.9652 \pm 0.0046$ | $0.9670 \pm 0.0041$ | $0.9660 \pm 0.0045$ | $0.9673 \pm 0.0040$ |
| | $wCDM$ | $0.9644 \pm 0.0048$ | $0.9655 \pm 0.0044$ | $0.9642 \pm 0.0047$ | $0.9650 \pm 0.0043$ |
| | $w(z)CDM$ | $0.9648 \pm 0.0046$ | $0.9631 \pm 0.0047$ | $0.9645 \pm 0.0048$ | $0.9634 \pm 0.0047$ |
| $w_1$ | $\Lambda CDM$ | $-$ | $-$ | $-$ | $-$ |
| | $wCDM$ | $-1.053 \pm 0.052$ | $-1.035 \pm 0.042$ | $-1.057 \pm 0.038$ | $-1.041 \pm 0.035$ |
| | $w(z)CDM$ | $-1.39 \pm 0.49$ | $-1.36 \pm 0.50$ | $-1.34 \pm 0.44$ | $-1.39 \pm 0.44$ |
| $w_2$ | $\Lambda CDM$ | $-$ | $-$ | $-$ | $-$ |
| | $wCDM$ | $-$ | $-$ | $-$ | $-$ |
| | $w(z)CDM$ | $-0.84 \pm 0.12$ | $-0.97 \pm 0.10$ | $-0.935 \pm 0.096$ | $-1.007 \pm 0.082$ |
| $w_3$ | $\Lambda CDM$ | $-$ | $-$ | $-$ | $-$ |
| | $wCDM$ | $-$ | $-$ | $-$ | $-$ |
| | $w(z)CDM$ | $-1.58^{+0.56}_{-0.50}$ | $-0.70^{+0.35}_{-0.29}$ | $-1.12^{+0.43}_{-0.38}$ | $-0.69 \pm 0.29$ |
| $w_4$ | $\Lambda CDM$ | $-$ | $-$ | $-$ | $-$ |
| | $wCDM$ | $-$ | $-$ | $-$ | $-$ |
| | $w(z)CDM$ | $< -1.03$ | $-1.91^{+0.76}_{-0.39}$ | $-2.6^{+1.6}_{-1.0}$ | $-1.81^{+0.66}_{-0.33}$ |
| $H_0$ | $\Lambda CDM$ | $67.37 \pm 0.62$ | $67.68 \pm 0.46$ | $67.51 \pm 0.61$ | $67.75 \pm 0.46$ |
| | $wCDM$ | $68.8 \pm 1.6$ | $68.4 \pm 1.0$ | $68.9 \pm 1.1$ | $68.59 \pm 0.86$ |
| | $w(z)CDM$ | $73.9 \pm 2.5$ | $69.2 \pm 1.5$ | $72.1^{+2.1}_{-1.8}$ | $69.6 \pm 1.3$ |
| $\sigma_8\Omega_m^{1/2}$ | $\Lambda CDM$ | $0.4656 \pm 0.0093$ | $0.4626 \pm 0.0087$ | $0.4646 \pm 0.0092$ | $0.4618 \pm 0.0084$ |
| | $wCDM$ | $0.4644 \pm 0.0098$ | $0.4633 \pm 0.0087$ | $0.4645 \pm 0.0095$ | $0.4637 \pm 0.0086$ |
| | $w(z)CDM$ | $0.456 \pm 0.010$ | $0.4661 \pm 0.0096$ | $0.4601 \pm 0.0098$ | $0.4650 \pm 0.0094$ |
| $y$ | $\Lambda CDM$ | $0.038^{+0.017}_{-0.021}$ | $0.039^{+0.017}_{-0.020}$ | $0.037^{+0.017}_{-0.019}$ | $0.037^{+0.017}_{-0.020}$ |
| | $wCDM$ | $0.041 \pm 0.019$ | $0.040 \pm 0.018$ | $0.044 \pm 0.019$ | $0.042 \pm 0.019$ |
| | $w(z)CDM$ | $0.042 \pm 0.019$ | $0.039^{+0.017}_{-0.021}$ | $0.045 \pm 0.019$ | $0.040^{+0.017}_{-0.020}$ |
| $R$ | $\Lambda CDM$ | $0.364 \pm 0.070$ | $0.363 \pm 0.071$ | $0.363 \pm 0.072$ | $0.367 \pm 0.071$ |
| | $wCDM$ | $0.362 \pm 0.070$ | $0.362 \pm 0.071$ | $0.359 \pm 0.071$ | $0.361 \pm 0.072$ |
| | $w(z)CDM$ | $0.359 \pm 0.070$ | $0.361 \pm 0.071$ | $0.356 \pm 0.071$ | $0.363 \pm 0.071$ |

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
