# Peer review of "CMB Tensions with Low-Redshift H0 and S8 Measurements: Impact of a Redshift-Dependent Type-Ia Supernovae Intrinsic Luminosity"

_symmetry, doi:10.3390/sym11080986_

Round 1
Reviewer 1 Report
Dear Sirs, the manuscript “ CMB tensions with low-redshift H0 and S8 measurements: impact of a redshift-dependent type-Ia supernovae intrinsic luminosity”, by Matteo Martinelli and Isaac Tutusaus investigates the H0 and S8 tensions, focusing on the possible effect that a SNIa intrinsic luminosity dependence on the redshift could have. They find that the tensions can be alleviated, however the inclusion of BAO data brings the tensions back. The motivation is good, the analysis has been performed in a correct way, and the paper contains publishable results. However, before I recommend publication the following minor issues should be addressed. 1. The fact that the tensions re-appear when BAO data are inserted, could it mean that there might be some unknown systematics in (some of) the BAO data? Can the tensions be removed if the authors use some instead of all BAO data? I do not ask to repeat the analysis, but just to add a comment if that could happen. 2. The authors could add a reference on 1810.05141 and 1903.10969 on how other dark energy Equation of State parapetrizations could alleviate the H0 tension. 3. The authors could mention that an alternative and interesting way to solve the tensions is through gravitational modification. Nevertheless, one should be careful in this case when he uses CMB data, since the latter assume intrinsically LambdaCDM at some steps. In summary, when the above points will be handled, I will recommend the paper for publication at Symmetry.
Author Response
First of all, we would like to thank the referee for his/her comments and suggestions that have definitely improved the content of our article. We appreciate that he/she has found our submission well motivated, the analysis correctly performed, and with publishable results. In the following we provide our reply to each one of the comments and we attach a revised version of our manuscript (please notice that this includes also edits done to satisfy the requirements of another referee).
The fact that the tensions re-appear when BAO data are inserted, could it mean that there might be some unknown systematics in (some of) the BAO data? Can the tensions be removed if the authors use some instead of all BAO data? I do not ask to repeat the analysis, but just to add a comment if that could happen.
We modified our comment on the effect of the BAO data in Section 5 (in red in this revised version) to account for the referee’s comment. We state explicitly that the removal of the SDSS DR12 data at the highest redshifts might help in easing the tension even with the inclusion of BAO. We comment that however, a detailed analysis of the BAO impact is needed.
The authors could add a reference on 1810.05141 and 1903.10969 on how other dark energy Equation of State parametrizations could alleviate the H0 tension.We thank the referee for pointing out these interesting references.
We were not aware of these works and we have now added these citations when we cite previous works that propose theoretical ways to alleviate the $H_0$ tension.
The authors could mention that an alternative and interesting way to solve the tensions is through gravitational modification. Nevertheless, one should be careful in this case when he uses CMB data, since the latter assume intrinsically LambdaCDM at some steps.
we mention the possibility of modified gravity models as a way out for the tensions discussed in Section 3, after we discuss our reconstruction methodology (in red in this revised version). We add some references to paper containing investigations of these models.We also comment on the complications that such models introduce in the use of some datasets, and we add references to papers discussing this point in more detail.

Reviewer 2 Report
In order to improve the readability and the reproducibility to the paper in question the following modifications are requested. 1) The word "tension" appears many times in the text but is defined only in formula (19).
I would suggest to insert actual formula (19) in the
introduction.
Also the abstract should contain a literal
definition of tension.
2) All the symbols of the paper should be defined
the first time they are used.
In particular the symbols of the LCDM cosmology should always
have the some nomenclature which are : Omega_m , Omega_lambda
and Omega_K, see as an example Hogg1999.
What means Omega_b and Omega_c in table I?
If they are different from the above three symbols they
should be defined.
3) What is the meaning of DE which characterizes the second column
of Table A1?
4) A comparison with the cosmological parameters
as given by the flat cosmology should be done,
see Zaninetti2019
------Bibliography--------------------
@ARTICLE{Hogg1999,
author = {{Hogg}, D.~W.},
title = "{Distance measures in cosmology}",
journal = {ArXiv Astrophysics e-prints},
eprint = {astro-ph/9905116},
keywords = {Astrophysics},
year = 1999,
month = may,
adsurl = {http://adsabs.harvard.edu/abs/1999astro.ph..5116H},
adsnote = {Provided by the SAO/NASA Astrophysics Data System}
}
@ARTICLE{Zaninetti2019,
author = {{Zaninetti}, Lorenzo},
title = "{A New Analytical Solution for the Distance Modulus in Flat Cosmology}",
journal = {International Journal of Astronomy and Astrophysics},
keywords = {Astrophysics - Cosmology and Nongalactic Astrophysics},
year = "2019",
month = "Jan",
volume = {9},
number = {1},
pages = {51-62},
doi = {10.4236/ijaa.2019.91005},
archivePrefix = {arXiv},
eprint = {1903.07121},
primaryClass = {astro-ph.CO},
adsurl = {https://ui.adsabs.harvard.edu/abs/2019IJAA....9...51Z},
adsnote = {Provided by the SAO/NASA Astrophysics Data System}
}
Author Response
First of all, we would like to thank the referee for his/her comments and suggestions that has helped improve the readability and reproducibility of our paper. In the following we provide our reply to each one of the comments. The attached file shows the revised manuscript with modifications in red, including also edits done to satisfy another referee.
The word "tension" appears many times in the text but is defined only in formula (19). I would suggest to insert actual formula (19) in the introduction. Also the abstract should contain a literal definition of tension.
We now include a literal definition of the tension in the abstract, as suggested by the referee. We prefer however not to have equations in our Introduction; we kept Eq.(19) in the methodology section where we define the method we follow for our analysis. We believe this is the appropriate place to introduce such a formula as we do not make use of it earlier; all the tensions values reported in the introduction have the appropriate reference to the papers where these were computed, which also include the formulas used for such calculations.
All the symbols of the paper should be defined the first time they are used. In particular the symbols of the LCDM cosmology should always have the some nomenclature which are : Omega_m , Omega_lambda and Omega_K, see as an example Hogg1999. What means Omega_b and Omega_c in table I? If they are different from the above three symbols they should be defined.
We appreciate the comment of the referee but we consider that all symbols are defined the first time they appear in the text. If the referee thinks that a specific symbol should be introduced in more detail we would appreciate a comment concerning the corresponding case.
Concerning the symbols of the LCDM model, we follow the nomenclature of the Planck Collaboration in their legacy articles. See Table 6 of https://arxiv.org/pdf/1807.06205.pdf where the authors claim that the minimal (flat) LCDM model is specified by the following 6 parameters: the physical baryon density $\Omega_b h^2$, the physical density of cold dark matter $\Omega_c h^2$, the amplitude $A_s$ and spectral index $n_s$ of the initial power spectrum, the angular scale of the acoustic oscillations $\theta_*$, and the optical depth to Thomson scattering from reionization $\tau$. We have marked in red in the text the lines were we describe the 6 parameters sampled of the LCDM model following Planck’s nomenclature, and we have the clarification that the minimal LCDM model accounts for a flat universe. Note that instead of sampling $\theta_*$ we choose to sample the Hubble constant $H_0$, but there is a unique relation between these two quantities once all the other parameters are fixed.
What is the meaning of DE which characterizes the second column of Table A1?
DE stands in the tables for Dark Energy. We have replaced this acronym (that was not defined in the paper) by Dark Energy in the tables.
A comparison with the cosmological parameters as given by the flat cosmology should be done, see Zaninetti2019
We appreciate the comment from the referee, but we consider that his comparison is already in the paper. In all the tables we provide the mean values and constraints for all cosmological parameters of the flat LCDM model, for different combinations of probes and datasets. Moreover, the flat LCDM posteriors appear in all triangular plots, and the comparison to the other models is discussed in the text. For clarification we have added a footnote the first time we refer to the flat LCDM model clarifying that we consider it as the concordance model in cosmology, and that we will refer to it just as LCDM in the following.
